# Assessment of the Response Profile to Hyaluronic Acid Plus Sorbitol Injection in Patients with Knee Osteoarthritis: Post-Hoc Analysis of a 6-Month Randomized Controlled Trial

**DOI:** 10.3390/biom11101498

**Published:** 2021-10-11

**Authors:** Olivier Bruyère, Germain Honvo, Eduard Vidovic, Bernard Cortet

**Affiliations:** 1WHO Collaborating Centre for Public Health Aspects of Musculo-Skeletal Health and Ageing, Division of Public Health, Epidemiology and Health Economics, University of Liège, 4000 Liège, Belgium; Germain.Honvo@uliege.be; 2Aptissen SA Medical Department, 1228 Plan-les-Ouates, Switzerland; e.vidovic@aptissen.com; 3Department of Rheumatology and UR 4490, University Hospital of Lille, 59000 Lille, France; Bernard.CORTET@chu-lille.fr

**Keywords:** hyaluronic acid, sorbitol, injection, osteoarthritis, post-hoc analysis

## Abstract

In a previous randomized trial, the non-inferiority of two hyaluronic acid injections (Synolis VA versus Synvisc-One) was assessed in patients with knee OA, with a response rate of 79% for Synolis VA. To assess whether a responder profile could be established for this treatment modality, we used the Synolis VA arm of a published 6-month prospective, multicenter, comparative, randomized, double-blinded trial. At baseline and during the study, pain and function were assessed using the Western Ontario and McMaster Universities Arthritis Index (WOMAC) questionnaire. Ninety-six subjects from the intention-to-treat trial were included in the analysis. The 6-month change of WOMAC Pain with Synolis VA was not associated with any baseline clinical data. However, the change in WOMAC Function was significantly associated with its baseline level, even after adjustment for potential confounding variables (*p* = 0.028), i.e., a poorer physical function at baseline was associated with a better response. In conclusion, in addition to the high absolute response rate to Synolis VA, the probability of success is even increased if administered in patients with more limited physical function at baseline. Further research with other potential confounding clinical variables is warranted in order to better applicate the concept of personalized medicine.

## 1. Introduction

The management of osteoarthritis (OA), the most prevalent form of arthritis, usually includes a combination of non-pharmacological and pharmacological modalities. Among the pharmacological treatments currently available, intra-articular hyaluronic acid (HA) injections play a substantial role in OA management, as highlighted in the updated version of the OA recommendations of two large international societies, namely, the European Society for Clinical and Economic Aspects of Osteoporosis and Osteoarthritis (ESCEO) and the Osteoarthritis Research Society International (OARSI) [1,2]. In addition, two groups of experts emphasized the moderate but significant efficacy of HA on OA symptoms, which is within the range of other pharmacologic OA treatment modalities [3]. More specifically, a meta-analysis highlighted that HA is efficacious by 4 weeks, reaches its peak effectiveness at 8 weeks, and provides a residual symptomatologic effect at 24 weeks [4]. At last, from data available from randomized controlled trials and summarized in different meta-analyses, HA is well tolerated and is not associated with any safety issue in the management of OA [5,6].

The current management practice is still largely based on the “one-size-fits-all” model, wherein patients diagnosed with the same condition are often prescribed the same treatment. However, since this method can lead to poor treatment response, a more patient-centric or personalized approach in medical OA practice is encouraged. In regards to HA, it was shown in a meta-analysis that all HA products could not be considered a homogeneous group, as there are differences in HA products that influence both efficacy and safety [7]. Interestingly, experts acknowledged that the variability of HA effects among different patient phenotypes has not been well understood [3]. Consequently, for a more personalized approach of OA management, there is a need for further investigation of patient characteristics associated with a better response to each particular HA treatment.

Recently, a 6-month prospective, multicenter, comparative, randomized, double-blinded trial was published comparing two HA products (i.e., one made of 80 mg hyaluronic acid and 160 mg sorbitol and the other made of 48 mg hylan GF-20) and their effects on pain and function efficacy in patients with moderate knee OA was published [8]. The objective of the present study is to investigate the responder profile to the HA made of sodium hyaluronate plus sorbitol. This study is called a post-hoc analysis as it is an additional and not pre-planned analysis of a published trial.

## 2. Materials and Methods

This study is a post-hoc analysis of data from a 24 week (168 days) prospective, randomized, phase IIIb, double-blind, controlled trial, which was designed to compare two different hyaluronic acid preparations for the symptomatic management of knee OA patients (non-inferiority design). The first compound was a solution containing 80 mg of hyaluronic acid and 160 mg of sorbitol, namely Synolis VA, and the second compound was made of 48 mg hylan GF-20 (Synvisc-One) [8]. Each of these compounds was administered as a single intra-articular injection. The current analysis only used data of the Synolis VA arm of the trial.

The original trial included patients of both genders, aged between 45 and 80 years, with radiologically confirmed knee OA, according to the American College of Rheumatology definition who were randomly assigned to one of the HA groups. The primary endpoint was the evolution of the Western Ontario and McMaster Universities Arthritis Index (WOMAC) for pain at day 168 following the injection, using the 100 mm Visual Analog Scale (VAS) rating. The WOMAC stiffness, function, and total scores assessed at day 168 were among the secondary endpoints in this trial. Patient response to treatment at day 168 (end of study) was also evaluated according to responder criteria proposed by the Osteoarthritis Research Society International (OARSI) Standing Committee for Clinical Trials Response Criteria Initiative and the Outcome Measures in Rheumatology (OMERACT), known as the OMERACT-OARSI set of responder criteria [9]. The OMERACT–OARSI criteria for response are (1) Improvement in VAS pain ≥50%, or (2) Improvement in at least 2 of the 3 following: VAS pain improvement from baseline ≥ 20%; Lequesne’s Algofunctional Index improvement from baseline ≥20%; Global Status Assessment (EQ-5D-5L) improvement from baseline ≥20%. Concomitant medications were allowed during the trial for pain relief when necessary for patient well-being, as long as they would not interfere with the investigational product. This could be prescribed by the investigator, but its use was to be kept to a minimum. The initial study trial was conducted in accordance with the ethics principals of the Declaration of Helsinki. It was approved and registered under no. 2017-A00034-49 to the ANSM, and ethical approval was obtained from CPP Ile-de-France.

### Statistical Analyses

The current post-hoc analysis was based on the intention-to-treat (ITT) population of the trial, which consisted of all subjects who received the Synolis VA injection (*n* = 96 patients) as this study used only data from the Synolis VA arm of the trial. Bivariate and multivariate regression analyses were used to assess baseline factors that could predict changes from baseline [(Day 0 score − Day 168 score/Day 0 score) × 100] in WOMAC pain and WOMAC function scores. Baseline factors predicting response to treatment at day 168, as assessed using the OMERACT-OARSI responder criteria (responders vs. non-responders), were evaluated by the means of logistic regression models (bivariate and multivariate). The model for improvement in WOMAC pain at day 168 included baseline WOMAC pain; whereby, for the model to predict improvement in WOMAC function at day 168, baseline WOMAC function was taken into account. Baseline WOMAC total score was considered as covariate in the multivariate model for response to treatment at day 168; afterwards, this covariate was replaced by WOMAC pain, function, and stiffness variables. Next, threshold values for baseline WOMAC pain, function, and stiffness for response to treatment at day 168 were determined by percentile analysis, in order to create WOMAC subscales binary threshold variables (below vs. above threshold). Once the percentile values were obtained for each variable, binary threshold variables (below vs. equal or above percentile value) were created for baseline WOMAC pain, function, and stiffness. Comparisons of the proportion of patients below versus above these thresholds, between responders and non-responders, were completed using the Fisher’s exact test. The WOMAC subscales binary threshold variables created were ultimately included in the multivariate models, instead of the original baseline WOMAC subscales specific variables. All the multivariate models were adjusted for potential confounding factors (i.e., age, sex, and BMI). The analyses were two-sided and *p* ≤ 0.05 was considered statistically significant. All statistical analyses were performed using the STATA software, version 14.2 (StataCorp LLC. College Station, TX, USA).

## 3. Results

### 3.1. Baseline Characteristics of Study Participants

The baseline demographic and clinical characteristics of study participants (PP population) were described in the original publication of the trial [8]. These characteristics are quite similar to those of the ITT population used in the current analysis. Briefly, 66% (65.63%) of patients in the whole ITT population were female, and the median age was 64.5 years (IQR: 58.0–72.0). More than two-thirds (68.75%) of these patients had a Kellgren–Lawrence OA grade of 3. For the 96 patients included in the present analysis, the mean WOMAC pain score at baseline was 46.68 ± 17.33 and the mean WOMAC function score was 43.12 ± 19.54.

### 3.2. Analysis of Factors Predicting Improvement in WOMAC Pain and Function and Response to Treatment, Considering Baseline WOMAC Subscales Variables as Predictors

The first analysis undertaken was to assess determinants predicting improvement in WOMAC pain at day 168, as compared to day 0. In bivariate and multivariable regression analyses, none of the covariates (i.e., age, sex, BMI, and baseline WOMAC pain) were found to be significantly associated with WOMAC pain change from baseline (data in file). For improvement in WOMAC function at day 168, baseline WOMAC function was the only predictor identified, both in bivariate analysis (*p* = 0.034) and in the multivariate model (*p* = 0.028) adjusted for age, gender, and baseline BMI (Table 1). The worse the WOMAC function was at baseline, the better the response to treatment.

Factors that predicted treatment response according to the OMERACT-OARSI criteria at the end of the study were then assessed. Firstly, we assessed whether baseline WOMAC total predicted response to treatment at day 168, and found significant associations; both in bivariate analysis (*p* = 0.003) and in the multivariate model (*p* = 0.002) adjusted for age, sex and baseline BMI (Table 2). In other words, for each increase of one point of WOMAC at baseline, the probability of being a responder was increased by 6%. None of the other covariates showed significant results.

Next, we searched for specific WOMAC subscales that determined the effect observed with baseline WOMAC total, by including the specific WOMAC subscales variables in the models, instead of baseline WOMAC total. In bivariate models, only baseline WOMAC pain (OR: 1.05; 95% CI: 1.02–1.09) and WOMAC function (OR: 1.05; 95% CI: 1.01–1.08) specific subscales variables were found to be significantly associated with treatment response at day 168; suggesting that baseline high levels of knee pain and worse function were significantly associated with positive response to treatment at day 168. However, no significant associations were observed in multivariate analysis including age, sex, and BMI, contrary to the result obtained for the multivariate model including baseline WOMAC total. Additional multivariate models were built, including separate specific WOMAC subscales (instead of all of these variables together), alongside the other baseline covariates (i.e., age, sex, and BMI). With these models, baseline WOMAC pain (*p* = 0.002) and baseline WOMAC function (*p* = 0.002) were found to be independently associated with response at day 168.

### 3.3. Determination of Threshold Values for Baseline WOMAC Subscales Scores for Being Responder to Treatment

Threshold values for baseline WOMAC pain, function, and stiffness scores for being a responder to treatment at the end of the study were then calculated. The main threshold values tested for response to treatment at day 168, using the Fisher’s exact test, were those corresponding to the 10th, the 25th, the 50th, and the 75th percentiles. From all the analyses conducted, the values corresponding to the 10th percentiles for baseline WOMAC pain, function, and stiffness scores, reached by 90% of the patients, were identified as the best discriminative values for responding to treatment at day 168. For baseline WOMAC function, the value corresponding to the 10th percentile was 18.76, showing significant difference between responders and non-responders to treatment at day 168 (*p* = 0.000). The value corresponding to the 10th percentile for baseline WOMAC pain was 26.8 (*p*-value comparing responders to non-responders equal to 0.018).

### 3.4. Analysis of Factors Predicting Improvement in WOMAC Pain and Function and Response to Treatment, Considering WOMAC Subscales Binary Threshold Variables as Predictors

The main analyses described in Section 3.2 were undertaken once more, this time considering the baseline WOMAC subscales binary threshold variables instead of the absolute values at baseline. As can be seen from Table 3 and Table 4, values yielded best regression coefficients and *p*-values (Table 3) and best ORs with significant 95% CI (Table 4) for the 10th percentile of baseline WOMAC function (WOMAC function binary threshold variable) as an independent predictor of change in function from baseline and of positive response to treatment at the end of the study.

## 4. Discussion

In the present study, we have shown that one independent determinant of symptomatic response to HA was the baseline level of the patient’s physical function limitation, i.e., the more important the limitation is, the more important the improvement will be. To the best of our knowledge, we are the first to suggest that the level of symptoms could impact the HA response in patients with OA. Interestingly, it should be noted that our results are consistent regardless of which definition of functional response to treatment is used (i.e., WOMAC function or OMARACT-OARSI criteria).

Our results could have significant impact for the management of OA in clinical practice. As a matter of fact, these results suggest that HA options should not be tried too early in the OA treatment process in order to maximize their effects. Consequently, we confirm HA should be used in patients with more advanced OA or in patients for whom initial pharmacological options were not sufficient or contraindicated. Interestingly, these data are consistent with the current version of the ESCEO guidelines for the management of OA [1]. Indeed, in this step-by-step approach of OA management, HA is proposed at the end of the second step, just after the use of symptomatic slow acting drugs in OA (SYSADOA), and topical and oral NSAIDs. Our results confirm that HA is well placed there, after testing of the first pharmacological modalities, such as when the patient is more severely symptomatic. At that time, chances of success from HA treatment are even more important.

With other treatment modalities, the phenotype profile of the patient who responds well could be different. For example, with the SYSADOA, a better response seems to be observed in patients with more recent OA or with less severe OA characteristics [10]. This is probably one of the reasons why SYSADOA are recommended in the first step of the ESCEO algorithm, i.e., in the early stage of the disease. With nonsteroidal anti-inflammatory drugs, other factors could be important, such as age, obesity status, and the presence of concomitant diseases (e.g., depression or diabetes mellitus) as was shown in a trial using rofecoxib, although rofecoxib is not used at present due to safety concerns [11]. Finally, it is noteworthy that the response to placebo has also been investigated. For example, in a meta-analysis, Zhang et al. noted that people with higher baseline pain were more likely to respond to placebo in OA [12].

The particularity of Synolis is that it contains sorbitol. High affinity between HA and sorbitol has been suggested to stabilize the complex through a very dense network of hydrogen bonds. The strong ability of sorbitol to scavenge and neutralize oxygen free radicals has been shown to delay degradation of the gel. Moreover, reducing concentration of free radicals may decrease migration of macrophages into the synovial membrane and reduce inflammation and pain [13,14]. HA plus sorbitol dose-dependently could suppress catabolic and inflammatory responses as well as oxidative stress-induced chondrocyte apoptosis in isolated human OA chondrocytes [15]. The suppression of these responses within joints may represent an important mechanism of clinical HA plus sorbitol action for OA treatment.

There are strengths and limitations in this study. We used a well-designed randomized controlled trial to investigate potential responders to HA. However, we are limited by the number of potential confounding variables used in the initial study, even if very few of them are well confirmed in the scientific literature. Consequently, some confounding variables with a potential larger impact may be more complex to obtain (i.e., genetic markers), and could have been missed even if the most important ones for the clinician have been assessed (i.e., BMI, age, sex, level of pain and function). One of the potential confounding factors potentially impacting OA response is compliance, however, this is not relevant here given the specific modalities of HA treatment (given once at baseline). Moreover, the responder profile observed in the present study could not be extrapolated to other HA products since it has been shown that HA efficacy varies widely across preparations [16]. It should also be pointed out that our population included mainly patients with mild to moderate OA and, consequently, our results could not be extrapolated to severe OA. At last, concomitant medications were allowed during the trial for pain relief but, given that their use was to be kept to a minimum, we do not believe that it has substantially influenced our results.

The future of personalized medicine will probably be different with the combination of a patient’s phenotype and the use of biological markers or metabolomics signature. For example, using a differential correlation network analysis method, Costello et al. identified a substantial number of metabolites for pain and function non-responders to total joint replacement, suggesting that inflammation, muscle breakdown, wound healing, and metabolic syndrome may all play roles in the response [17]. Another example, but much more specific, was shown using autologous chondrocyte implementation treatment whereby different biologically relevant protein changes were associated with the response, suggesting that several pathways appear to be altered in non-responders [18]. A last example is the use of serum level of lysophosphatidylcholines to phosphatidylcholines ratio that has been shown to be associated to the response to licofelone and naproxen in patients with knee OA [19].

## 5. Conclusions

In conclusion, a more altered physical function seems to be associated with more important function relief when using HA and particularly Synolis. More specifically, having a WOMAC function score over a value corresponding to the 10th percentile of baseline WOMAC function (value = 18.76 in the current study) appears to be the best predictor of treatment response at day 168, and of change in function from baseline. It confirms the usefulness of viscosupplementation in patients for whom oral treatment with SYSADOAS, NSAIDs, or other analgesics provide insufficient clinical responses or are poorly tolerated. Further research with a larger sample sizes are needed to confirm the preliminary results obtained in this study, in order to help to personalize the treatment of OA for the best satisfaction of the patient.

## Figures and Tables

**Table 1 biomolecules-11-01498-t001:** Factors predicting improvement in WOMAC function after 168 days.

Factors	Bivariate Model			Multivariate Model (*n* = 91)		
	β	SE	*p*	β	SE	*p*
Age	–0.49	0.61	0.420	−0.39	0.61	0.521
Sex (female)	–9.78	12.43	0.434	−13.69	12.68	0.283
BMI	1.18	1.35	0.385	0.75	1.34	0.576
Baseline WOMAC function	0.65	0.30	0.034	0.70	0.31	0.028

**Table 2 biomolecules-11-01498-t002:** Factors predicting OMERECT/OARSI response to treatment at day 168, considering baseline WOMAC total as predictor.

Factors	Bivariate Model		Multivariate Model (*n* = 93)	
	OR	95% CI	OR	95% CI
Age	1.02	0.97–1.07	1.03	0.97–1.08
Sex (female)	0.66	0.23–1.90	0.36	0.11–1.21
BMI	1.03	0.92–1.15	1.01	0.90–1.13
Baseline WOMAC total	1.05	1.02–1.09	1.06	1.02–1.09

**Table 3 biomolecules-11-01498-t003:** Factors predicting improvement in WOMAC function at the end of the study as compared to baseline (day 0), including baseline 10th percentile (P10) of WOMAC function as covariate.

Factors	Multivariate Model for Improvement in WOMAC Function (*n* = 91)		
	β	SE	*p*
Age	−0.65	0.59	0.276
Sex (female)	−6.59	12.02	0.585
BMI	1.20	1.29	0.355
P10 baseline WOMAC function	63.97	19.02	0.001

**Table 4 biomolecules-11-01498-t004:** Factors predicting response to treatment at the end of the study—model including baseline WOMAC subscales binary threshold variables as covariates.

Factors	Bivariate Model for Response to Treatment According to the OMERACT-OARSI Criteria (*n* = 93)	
	OR	95% CI
Age	1.02	0.96–1.08
Sex (female)	0.57	0.16–1.95
BMI	1.02	0.89–1.16
P10 baseline WOMAC pain	3.51	0.70–17.47
P10 baseline WOMAC function	13.81	1.23–155.29
P10 baseline WOMAC stiffness	0.75	0.06–9.19

## Data Availability

Not applicable.

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
