# Peer review of "Assessment of the Response Profile to Hyaluronic Acid Plus Sorbitol Injection in Patients with Knee Osteoarthritis: Post-Hoc Analysis of a 6-Month Randomized Controlled Trial"

_biomolecules, 2021, doi:10.3390/biom11101498_

Round 1

Reviewer 1 Report

Review Assessment of the response profile to hyaluronic acid plus sor- 2 bitol injection in patients with knee osteoarthritis: post-hoc 3 analysis of a 6-month randomized controlled trial

Comments

  1. The paper is well organized and well written. The final result is very interesting and important for practical use. However, significant association of baseline WOMAC pain index with the drug efficacy should be also indicated in the Conclusion.
  2. This article shows that the intraarticular administration of Hialuronic acid (HA) in some patients with moderate Osteoarthritis ( maximum grade 3 ) present better results when the illness is more advanced.3
  3. In present the indications of using intraarticular HA are for moderate OA which means that actual study does not provide much novelty in the field.
    This article sustains the well-known idea that this treatment works better in this stage, this can be considerate a strength of this article but not a novelty. A strength because reiterate and consolidate the idea that HA is efficient in moderate OA.
    4. The fact that patient received medication for pain relief can modify the results obteind.
    ‘Concomitant medications were allowed during the trial for pain relief, when necessary for patient well-being and would not interfere with the investigational product. This could be prescribed by the investigator, but its use was to be kept to a minimum.’
    5. The statistical analysis and the results are hard to read and to understand. This article is addressing to doctors or patients and must be clear and concise. My opinion is that they need revision to be clearer.

Author Response

Reviewer 1

  1. The paper is well organized and well written. The final result is very interesting and important for practical use. However, significant association of baseline WOMAC pain index with the drug efficacy should be also indicated in the Conclusion.

Authors: We acknowledge that our text was not clear enough and we have modified our text to say that pain was not associated with drug efficacy. We are sorry for the confusion.

  1. This article shows that the intraarticular administration of Hialuronic acid (HA) in some patients with moderate Osteoarthritis ( maximum grade 3 ) present better results when the illness is more advanced.

Authors: we agree and it has been highlighted that we did not included to early OA but also too severe OA

  1. In present the indications of using intraarticular HA are for moderate OA which means that actual study does not provide much novelty in the field. This article sustains the well-known idea that this treatment works better in this stage, this can be considerate a strength of this article but not a novelty. A strength because reiterate and consolidate the idea that HA is efficient in moderate OA.

Authors: we thank the reviewer for highlighted our most important results

  1. The fact that patient received medication for pain relief can modify the results obteind.
    ‘Concomitant medications were allowed during the trial for pain relief, when necessary for patient well-being and would not interfere with the investigational product. This could be prescribed by the investigator, but its use was to be kept to a minimum.’

Authors: We agree that it is a very important point that is now pointed out in our discussion section.

  1. The statistical analysis and the results are hard to read and to understand. This article is addressing to doctors or patients and must be clear and concise. My opinion is that they need revision to be clearer.

Authors: we respectfully suggest not to modify to much the method section.  Indeed, we strongly believe that it is important for the readers to have an exhaustive view of the methodology.     However, we have made the abstract much more simple and clearer.

Reviewer 2 Report

This is an interesting post-hoc analysis on factors influencing the response rate after an intraarticular knee HA+sorbitol injection compared to HA only. The manuscript is well written, the methodology is strong and the results may be useful in the selection of new tools to treat moderate OA.

Some minor comments:

  • The abstract length should be significantly reduced. Actual words are 362 versus the maximum 200 as per Instruction for Authors. Moreover, no headings should be mentioned.
  • 30% of references are self-citations. Please choose only 2 pertinent manuscripts.
  • Please mention the extended form of OMREACT-OARSI and WOMAC before using the abbreviated one in the abstract.
  • Please adjust spacing throughout the text.
  • Authors should better explain the biochemical difference between the two HAs and the additional role of sorbitol in Synolis VA.
  • It is not clear how the sample which underwent the post-hoc analysis was extracted by the original cohort. Authors should specify it.

Author Response

Reviewer 2

  1. The abstract length should be significantly reduced. Actual words are 362 versus the maximum 200 as per Instruction for Authors. Moreover, no headings should be mentioned.

Authors : we fully agree with the reviewer and our abstract has been modified

  1. 30% of references are self-citations. Please choose only 2 pertinent manuscripts.

Authors: we agree and we have removed 5 of the citations and added 3 new ones.

  1. Please mention the extended form of OMREACT-OARSI and WOMAC before using the abbreviated one in the abstract.

Authors: it has been done

  1. Please adjust spacing throughout the text.

Authors: done.

  1. Authors should better explain the biochemical difference between the two HAs and the additional role of sorbitol in Synolis VA.

Authors: it is an important comment and we have provided some information in the text.

  1. It is not clear how the sample which underwent the post-hoc analysis was extracted by the original cohort. Authors should specify it.

Authors: we agree it was not clear and we have modified our text. As a matter of fact, we just used the one of the two arms of a previous non-inferiority trial.